# Research Series Review for Transdisciplinarity Assessment—Validation with Sustainable Consumption and Production Research

**Tomohiko Sakao** 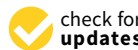

Division of Environmental Technology Management, Department of Management and Engineering, Linköping University, 58183 Linköping, Sweden; tomohiko.sakao@liu.se; Tel.: +46-73-620-9472

**Abstract:** In light of the escalating challenges for the sustainability of our societies, the need for improving the research of sustainable consumption and production (SCP) aiming to make real changes on the societies towards sustainability is evident. Transdisciplinary (TD) research is a promising way to enhance SCP research; however, insights to operationalize the concept of TD research are needed for both funders and researchers. Therefore, this article proposes an innovative way to capture and analyse a research series for transdisciplinarity assessment in qualitative and quantitative terms. This new way is termed research series review (RSR). This article adopted literature analysis and partly reflexive retrospective reasoning. In particular, citation content analysis was carried out in relation to two research series selected as the cases. The results show that RSR has advantages such as clearer traceability with cause-and-effect relationships. Furthermore, a successful SCP research series is hypothesised to form an iterative process between practical and theoretical fields as well as finding opportunities and proposing solutions.

**Keywords:** product/service system (PSS); circular economy; transdisciplinary research; engineering design; research evaluation; co-creation; research series review (RSR); causal impact; honeybee dancing with inquiry

## 1. Introduction

Our societies face escalating, grand challenges such as those involving sustainability and aged populations. Environmentally sustainable consumption and production (SCP) [1] is an urgent and important issue among others. It is, in fact, recognized as important globally and highlighted as Goal 12 of the UN's SDGs (United Nations' Sustainable Development Goals), that is, to "ensure sustainable consumption and production patterns". SCP can be defined as "the use of services and related products which respond to basic needs and bring a better quality of life while minimising the use of natural resources and toxic materials as well as the emission of waste and pollutants over the life cycle of the service or product so as not to jeopardise the needs of future generations" [1]. It also serves as a basis for facilitating the transition of societies towards a circular economy (CE), which is "one that is restorative by design, and which aims to keep products, components and materials at their highest utility and value, at all times" according to Webster [2]. SCP is clearly based on societal needs, and therefore, SCP research (e.g., [3,4]) should be focussed upon solving the societal challenges. But, facing limited diffusion of SCP in our societies thus far (see e.g., [5]), the need for improving SCP research aiming to make real changes in societies towards sustainability is evident.

To make changes in societies, it is already highly effective in research planning to involve actors such as industry practitioners and policy makers that will play important roles for the expected changes. In addition, to address the real-world problems effectively and efficiently, research addressing

multiple academic disciplines simultaneously is especially needed [6]. Nowadays, in fact, more transdisciplinary (TD) research, for which involving stakeholders and addressing multiple disciplines are the two common features [7], is increasingly demanded by research funders [8]. According to Rosenfield [9], which concerns human health and well-being as a societal challenge, TD "projects are those in which researchers from different fields not only work closely together on a common problem over an extended period but also create a shared conceptual model of the problem that integrates and transcends each of their separate disciplinary perspectives." In the research realm of sustainability science, there seems to be an agreement that sustainability challenges require TD research [10]: a conceptual framework to analyse TD research is proposed and applied to energy transition [11], and a reference framework is adopted for TD research addressing socio-environmental systems [12]. TD research can be a promising way to gain insights to enhance implementation of SCP, which tackles societal problems often requiring multiple disciplines, as indicated by an advanced work [13]. However, a practical yet science-based procedure for assessing research planned or performed from the TD research perspective is still debated [14–16].

TD research is ideally carried out with three phases according to [10]: collaborative problem framing, co-creation of transferable knowledge, and integrating and applying the co-created knowledge. In particular, synthesis process is regarded as important even in the planning [17]. The TD research process practically involves stakeholder engagement, problem sharing, solution development by academics of multiple disciplines and solution implementation that together take a long time [17]; this differs from single disciplinary research, which does not demand many elements of this process. TD research thereby tends to meet specific challenges such as discontinuous participation and tracking scientific and societal impacts [10]. Therefore, in order to be successful, TD research requires grand design of research addressing multiple disciplines in a synthetic manner and showing the intended impacts on academia and societies. Assessment limiting to a piece of research work, e.g., in a one-year project plan and a single publication, risks overlooking important aspects of the entire TD research. Therefore, in the TD context, it is crucial to set an assessment unit as a series of research works and to address intended impacts. Therefore, this article aims to propose a new systematic manner to capture and analyse a series of research works from the TD perspective. To embrace this way of systematic analysis, a new term, research series review (RSR), is coined. In order to verify and validate this systematic manner, materials could be sourced from a pool of project plans or scientific publications. The latter is adopted in this article because of the higher accessibility. The guiding research questions (RQs) are set out as follows.

1. How can one capture a series of research works from scientific publications?
2. What are relevant indicators to assess a series of research works from the TD perspective?

Insights to operationalize the concept of TD research are important and needed for researchers and funders to strategically plan, facilitate and evaluate more effective and efficient research from the TD perspective. Virtually no method available in literature allows the systematic capture of such a unit of research work for transdisciplinary assessment. The insights to be obtained are expected to increase the quality of SCP research and thereby its impact on societies, because involving relevant stakeholders throughout its process is required for realizing SCP efficiently and intended by TD research. This contribution will be significant, especially by improving the strategic design of a series of research works for SCP.

## 2. Method and Materials

### 2.1. Research Series Review (RSR)—the Proposed Method

What is suggested to be called RSR follows the concept and merit of a systematic literature review [18] in case it is applied to scientific publications. The purpose of RSR is to capture a series of research works that have cause-and-effect relationships and to get an understanding of the series from

a helicopter view. RSR builds upon the idea that a research work could be modelled as a collection of genes, using an analogy partly in line with Ref. [19]. A gene here means a chunk of valuable contribution to a research work from the viewpoint of scientific research, and is either created by the publication as such or inherited from cited publications. This implies that every citation does not belong to this inheritance, because citing a publication functions in various manners such as to merely acknowledge earlier related works with a certain subject without really using the specific knowledge of the publication. The idea of the gene is the backbone for a genealogy of research works, representing a research series, and enables review of publications focusing on cause-and-effect relationships. RSR is in theory applicable to a series of research works that are under planning as well as were performed. The four steps of RSR with scientific publications are explained below.

1.  Identify an article and its genes

A published research article is identified according to the objective of the particular review. Its genes are selected partly depending on the focus of the review. A gene is typically presented as a scientific contribution or practical implication of an article.

2.  Arrange articles in generations

A database of literature that includes citation relations is selected; databases such as Web of Science and Scopus have this information. Then, articles of citation relations with the concerned article are found from the database and arranged in such a way that the article is centred with preceding (parent) and following (child) generations of articles. A generation here means a group of articles that have a relation to the article in focus through a certain number of rounds of citation. The number of generations addressed by a specific RSR needs to be decided in both directions in advance (discussing how many generations are suitable for RSR is beyond this article). For instance, an article in the second following generation cites another article in the first following generation, which cites the article in focus, as depicted by the generations in Figure 1. Similar to the ordinary meaning of a generation for people in a society, individuals in a generation do not always carry an identical gene. Note that, in this arrangement, an article may be located in multiple generations; for example, Article A is cited by Article B and Article C, which is cited by Article B (as depicted by "Given relations for citation" in Figure 2); Article B appears in two generations ("Arrangement in generations" in Figure 2). This step is carried out solely on citation relations and therefore is performed mechanically.

3.  Capture a genealogy centred by the article

From the arrangement made by the previous step, captured is a genealogy containing only the articles that carry a gene of the article in focus, resulting in a set of articles for each generation (the first child and parent generations and so forth). Therefore, this step requires content analysis, and focuses on semantic analysis instead of a syntactic one [20]. Among the four types of function of citation (ibid.), that is, 1. provide background information, 2. construct theoretical framework, 3. provide previous empirical/experimental evidence, and 4. describe challenges, the first one needs to be excluded. With regard to disposition of citation (ibid.), positive disposition needs to be found in the citation. For instance, in a genealogy, an article in the second following generation improves a solution proposed by another article in the first following generation that tackles a problem formulated by the article in focus. While Step 2 is nearly automatic thanks to the database, this step requires serious work by an investigator. However, this is considered to be the best way, and this manual work may be, at least partially, inevitable in order to capture cause-and-effect relationships in the literature. Note that those in the second and further generations in a genealogy also have a substantial relation to the article in focus, which makes the genealogy rooted to the focal article sensible; however, an article substantially using knowledge of the article in focus, for instance, often cites the article, thus is in the first following generation. Obviously, articles in a genealogy are not necessarily written by the authors of the article in focus. In the example in Figure 1, the articles with blue-shaded nodes are captured together, with the links between them maintained.

A complex situation with regard to citation relations may occur, as exemplified by Figure 2. Different strategies exist, and one suitable to the objective of the RSR needs to be adopted. In the case of prioritizing comprehensiveness of the genealogy, Alternative 1 may be taken. For simplicity, Alternatives 2 or 3 may be chosen; the strongest relation is nominated in this case.

4.　Derive a narrative for the genealogy

A narrative is derived to document how the concerned series of research works evolved with cause-and-effect relations. This step functions as a synthesis of articles in a genealogy. Note that this step is also performed only based on the contents of the articles in a genealogy.

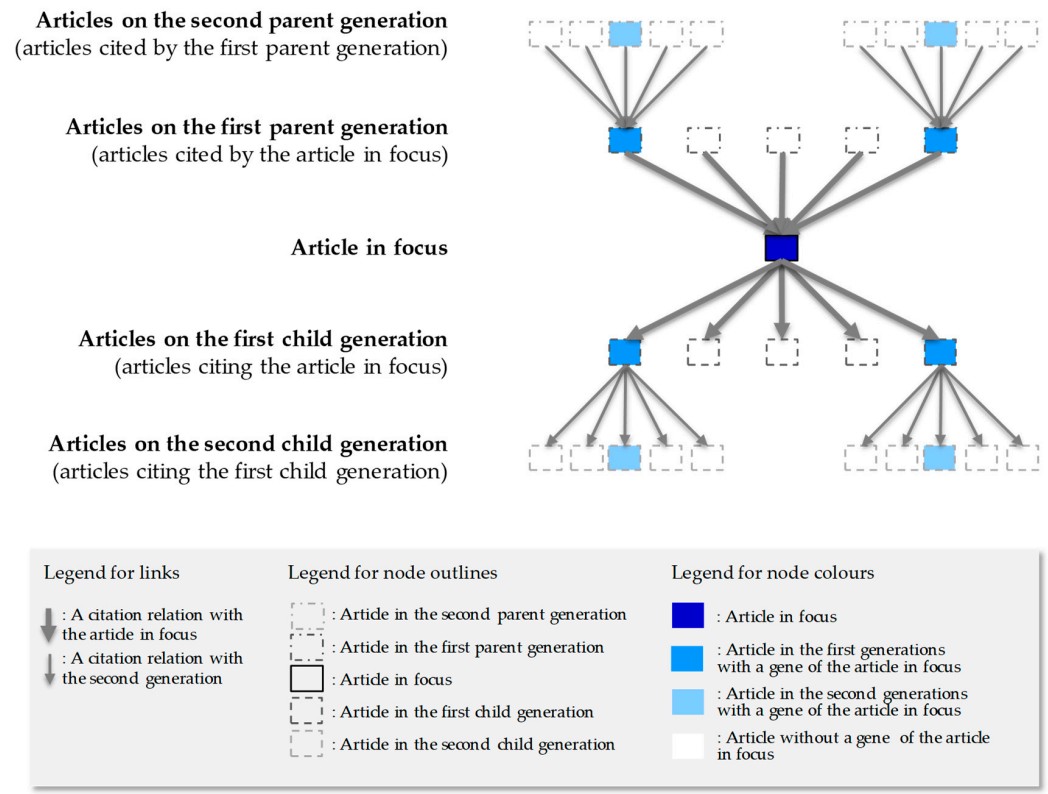

**Figure 1.** Schematic representation of the literature arranged in multiple generations. Note: The preceding and following generations are termed as parent and child generations, respectively, for intuitive understanding. This schematic representation depicts only the first and second generations, but the generations in a scope may continue according to the relations found.

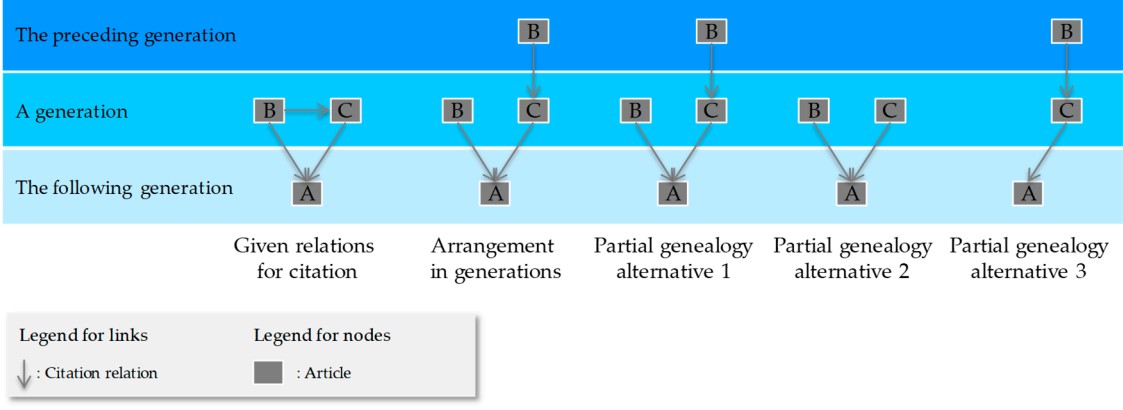

**Figure 2.** Different alternatives for forming a partial genealogy of the given citation relations.

*2.2. Materials*

The database for the literature chosen in this research was Scopus, because it is among those containing the most publications and is thus suitable for more comprehensive RSR. Journal articles written in English in the database were selected for the procedure. Two series of research were targeted for this RSR for verification and validation of RSR; one from SCP research (a focus of this article). The article in focus is Ref. [21] in product/service system(PSS) design research from the engineering design realm, published in 2007, because it is arguably one of the most known (cited 236 times in Scopus). An additional reason for this choice is to enrich the discussion of implications for PSS design research and SCP research at large using the author's own knowledge and experience about Ref. [21] (this part corresponds to reflexive retrospective reasoning performed by, e.g., [22]). The motivations for the choice of PSS within SCP are as follows. First, regarding SCP (as defined in Section 1), PSS is arguably one of the most central research objects for SCP, as it could be defined as "a system of products, services, supporting networks and infrastructure that is designed to be: competitive, satisfy customer needs and have a lower environmental impact than traditional business models." [23]. One of the origins of PSS research is SCP as shown by, e.g., [21,23] and PSS has been heralded as one of the most effective instruments for SCP according to a review [24]. Second, research of PSS in the last decade or so encountered barriers in the SCP and TD contexts; although substantial works such as an inter-disciplinary research work addressing engineering design and marketing to derive insights and recommendations to facilitate industry take advantage of PSSs [25] were performed, a critique is found from the standpoint of pursuing environmental contributions against research published after 2006 [24]—PSS research mainly uses case study research and simply seems to confirm the findings of the pre-2006 literature [24]. Concerning PSS implementation, despite some research progress to shed light on this issue, diffusion of PSSs in our societies is thus far limited [26] (see recent findings on these issues, e.g., [27,28]).

The other series is taken from robust design research. Robust design research aims to increase the robustness of a product by reducing its variation in functional performance that has a negative effect on the reliability and perceived quality of a product [29]. It is rooted in Taguchi's seminal works (e.g., [30]) and contributes to designers determining design variable settings to optimize product performance against multiple criteria [31]. It often adopts the mathematical formulation of problems to derive solutions. It was chosen in this research because it is also in the engineering design realm (meaning similar values about publication) but is expected to be less TD research than PSS design research. The specific selection criteria for the other article in focus were as follows:

- a journal article registered in Scopus (the chosen database),
- concerning robust design using optimization (due to the above-mentioned reason),
- having a similar degree of academic impact as Ref. [21], and
- having a similar age as Ref. [21].

The last two criteria were introduced to make comparison between the two research series sensible based on the presumably similar numbers of articles in the two research series and the similar publication trends (e.g., numbers of references on a paper may change substantially over the years—see Section 5.2.1.) at the time. The article that matches the criteria above most according to the title and abstract was found to be Ref. [32]; the query and search hits are shown in Appendix A (the author did read the titles and abstracts of these publications). It should be noted that these two articles are examples used for testing RSR and there is no intention to argue Ref. [21] as a representation of SCP research. In addition, two series are deemed sufficient for the comparison. The search results with Scopus shown in this article are all valid as of July 2019.

## 3. Results

### 3.1. Overview

Following the procedure described in Section 2.1., RSR was performed with the materials explained in Section 2.2. Namely, for each of the two articles, an arrangement was made, then a genealogy was captured, and finally a narrative was derived. This RSR was performed against the whole given population except articles with full-text unavailability. The journals investigated are listed in Appendix B. All the articles found in the arrangement are listed in Supplementary Tables S1 and S2 in the supplementary materials uploaded. The search result in this article was confirmed in July, 2019. Each result with details for the two articles will be presented in Sections 3.2 and 3.3, respectively.

The genealogy for each article was found to include 12 and 17 articles in total for [21] and [32] (including itself), respectively, as shown in Table 1. Articles cited by the article in focus within the population (i.e., English articles registered in Scopus) (13 articles for each article in focus) were considered to identify two articles for each in the first parent generation of the genealogy. Those in the second parent generation were searched from those cited by the two articles per each in the first parent generation (8 and 28 articles for [21] and [32], respectively). Articles citing the article in focus (97 and 139 for [21] and [32], respectively) were investigated to identify articles in the first child generation of the genealogy. Articles citing the first child generation of the genealogy (95 and 126 articles for [21] and [32], respectively) were investigated to identify articles in the second child generation. The total of the numbers in Table 1 is 546, indicating the number of investigations of citation content analysis (besides the two articles in focus). This means that the accuracy of semantic analysis for citation explained in Section 2.1. was ensured for the 546 articles and not each article was entirely read.

**Table 1.** Numbers of articles in the generations for the two articles.

|  | Genealogy of Ref. [21] | Investigated for Ref. [21] | Genealogy of Ref. [32] | Investigated for Ref. [32] |
|---|---|---|---|---|
| The Second Parent Generation | 0 | 8 | 0 | 28 |
| The First Parent Generation | 2 | 13 | 2 | 13 |
| The First Child Generation | 4 | 97 | 12 | 139 |
| The Second Child Generation | 5 | 95 | 2 | 126 |

The numbers of articles in the genealogy for [21] and [32] evolved over the years as shown in Figures 3 and 4, respectively. These graphs show many years taking effects from one article to another in the genealogies.

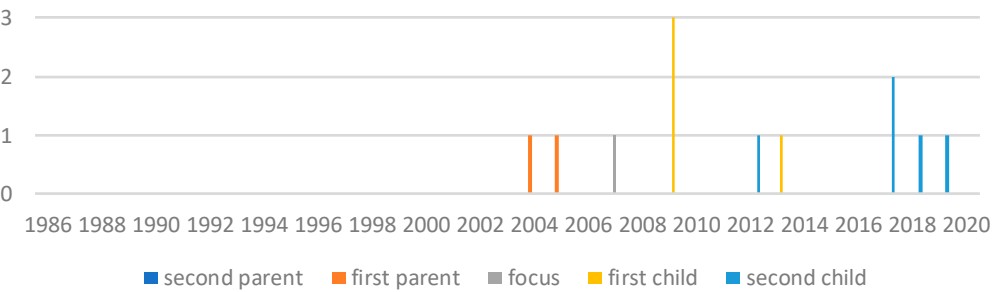

**Figure 3.** Number of articles over the years in the genealogy for Ref. [21].

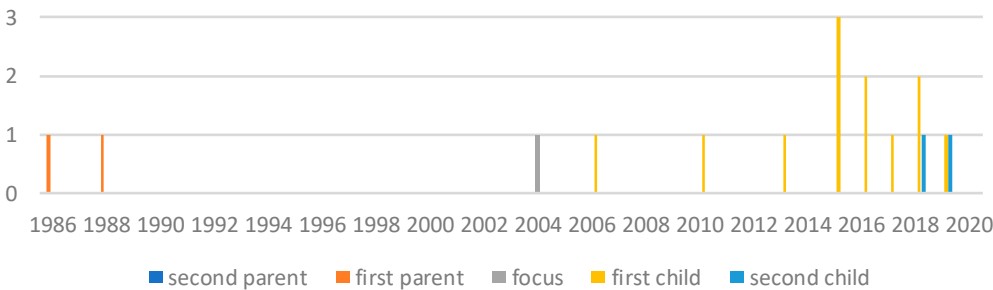

**Figure 4.** Number of articles over the years in the genealogy for Ref. [32].

### 3.2. Genealogy for PSS Design Research

This paper [21], titled "Service Engineering: a novel engineering discipline for producers to increase value combining service and product", was published in 2007 in the Journal of Cleaner Production. Exact phrases are quoted as much as possible for the narrative to reduce the risk of the bias of the author of this presented article. According to the content of Ref. [21], "[i]n the context of eco-design", "a much bigger framework" was needed "to pursue sustainability". "This call[ed] for establishing a new discipline". Therefore, "[t]his paper aim[ed] at proposing a novel engineering discipline for producers toward sustainable production and consumption, service engineering (SE)." "[A] methodology of modelling and designing services, and a computer-aided design tool called Service Explorer, are presented." Then, they are shown "to be effective through two applications." A major advantage of SE is to "allow(s) designing services in parallel with products", corresponding to exchangeability [33]—the core of PSS design.

According to the content of Ref. [21], this paper's genes created by itself may be chosen as the design method, the computer-aided design tool, and the two applications. The method is described as a procedure and also presented as a flow chart. One application of the method and the tool was realized for and in collaboration with a hotel in Italy: "the authors carried out an on-site customer survey about guests' requirements." In addition, they "investigated the actual hotel service by interviewing the employees and the hotel owner". After the application, feedback on the solutions from the design method and tool was "obtained from the hotel company" regarding the newness and feasibility to show the effectiveness of the method and tool.

These genes are shown to be partly inherited from [34,35]. Ref. [34] contributes to part of the modelling method and the design tool, while Ref. [35] provides part of the design method and a portion of the application to the hotel service. Therefore, Ref. [34] and Ref. [35] are both recognized in the first preceding generation in the genealogy. No article was recognized in the second preceding generation. The entire genealogy is depicted in Figure 5.

In the other direction, the genes of Ref. [21] have been used in the following generations. One article [36], which "developed some working steps based on . . . and the service evaluation method [21]" (the reference labels in all the quotes are replaced with those used in this article) and applied it with the tool in "a German manufacturing company", is recognized in the first following generation in the genealogy. Several articles citing [36] are recognized in the second following generation: two references [37,38] developed the extended method [36] further through application to manufacturers of production machines and washing machines, respectively. Ref. [39] reports a case where an "industrial provider of investment machinery" "applied a method for PSS design" [36]. From the TD research perspective, this research differs substantially from its preceding research [21,36–38], because the subject (i.e., the problem owner) shifted from researchers to practitioners.

Without interacting with practitioners substantially, Ref. [40] compared the extended method [36] with other design methods with commonality to "give hints to designers in applying these" methods. A similar comparison was performed for functional modelling across disciplines by [41] to gain

insights that "lead to the identification of specific needs and opportunities". It states "[e]xcept for [21], none of the reviewed PSS design approaches was found to propose a sequential functional modelling approach" and thus acknowledges the uniqueness of Ref. [21] in PSS design research. This comparison was then extended to "empirical studies in ten companies developing mechatronic systems and/or product-service systems" in Ref. [42], deriving "future research endeavours pertaining to the development of support for collaborative, (cross-)disciplinary function modelling."

Several articles were excluded from the genealogy because of insufficient use of the genes. For instance, Ref. [43] states "Service Explorer has been verified through several cases in various industries so far; ... and a hotel in the accommodation industry [21]. Verifications are now being conducted in Sweden, Germany, Denmark, and Japan", but gene usage is unclear. Similarly, Ref. [44] citing Ref. [21] states "how this new business by means of integration of products and services is achieved" in industry through a survey is interesting because it researches practice to identify a research opportunity by saying "there do not seem to be any established methods or tools developed to support the development" of PSSs. However, no gene is clearly presented.

### 3.3. Genealogy for Robust Design Research

This paper [32], titled "Robust design of structures using optimization methods", was published in 2004 in a journal called Computer Methods in Applied Mechanics and Engineering. To reduce the risk of the bias of the author, a similar style as used for Ref. [21] above is adopted for the narrative below as much as possible. According to Ref. [32], "in the design of engineering structures for improving the structural performance and reducing costs", "methods of structural optimization" are important. Therefore, "robust design of structures with stochastic parameters is studied using optimization techniques" and "[t]he robust design of structures is formulated as a multicriteria optimization problem." A method incorporating "the second-order perturbation method" is proposed, to enable an engineer to select the parameter set to the best compromise for the design: the primary solution and the gradients entering the algorithm are obtained by a Taylor-series approximation to the second order. Then, "[t]he applicability of the proposed method has been demonstrated by numerical examples." The proposed method "is advantageous in terms of computational costs compared with Monte Carlo simulations".

According to the content of Ref. [32], this paper's genes created by itself may be chosen as the proposed method and four defined problem examples. The method is described with a large number of mathematical formulae. Two of the problem examples are here depicted by Figure 6. The problem to be solved with Figure 6a is described as "[t]he structural compliance" of the truss structure "resembling a power transmission tower is to be minimized." That with Figure 6b is "to minimize the peak value of the vertical displacement response of the fifth node of the planar 10-bar structure". The problems are represented in an abstract manner but can be understood to be relevant to real-world design problems even by a layman: one can see objects with similar structures as Figure 6a in our societies and envision Figure 6b as frequently addressed by designers of physical structures. These problems were solved in Ref. [32] by running computer software to show advantages such as short computation time.

These genes of Ref. [32] are shown to be partly inherited from Ref. [45] and Ref. [46]. Both contribute to part of the second-order perturbation method. Ref. [45] especially concerns the probabilistic finite element method, while Ref. [46] addresses transient analysis in particular. Therefore, Ref. [45] and Ref. [46] both may be recognized in the first preceding generation in the genealogy. They adopt a similar research style proposing a method mathematically formulated that solves a problem via computation. Notably, they were published nearly two decades before the article in focus. Ref. [45] and Ref. [46] have 454 and 80 citations, respectively, and thus may be considered impactful in general. No article was recognized in the second preceding generation in the genealogy. The entire genealogy is depicted in Figure 7.

In the other direction, the genes of Ref. [32] have been used in the following generations. Twelve articles may be recognized in the first following generation in the genealogy. One of the four problems

defined by Ref. [32] has been used at least partly by seven articles, while part of the method has been used by six articles (one article used part of the problem and the method both). Among others, in approaching the real-world complexity, the method [32] was extended to large deformation processes of solids as encountered in metal forming [47]. Interestingly, Ref. [48], after applying its original method for structural modelling to an actual industrial structure, identified a research opportunity to develop a model that "is satiable for robust design" referring to Ref. [32]. In this case, the use of any gene of Ref. [32] was not recognized; however, this way of development was unique in relation to Ref. [32]. Next, in the second following generation in the genealogy, two articles may be recognized. One of the two [49] borrows a part of the method by stating "a previously developed single objective function [50], borrowed from the field of robust design [32], is used" ([50] is in the first following generation). The other uses one of the truss problems and cites one of the twelve in the first following generation, however, ought to cite Ref. [32] directly.

### 3.4. Analysis of Results from TD Research Perspective

#### 3.4.1. Overview

The results of RSR with the two series of research works are analysed from the TD research perspective. As already reviewed in Section 1, involving stakeholders and addressing multiple disciplines are the two common features for TD research [7] in line with Ref. [9]. In addition, the outcome knowledge provided, i.e., a problem (challenge) or solution, is especially relevant to TD research, which aims to solve a problem of a society [8], addressed by, e.g., Ref. [51]. Therefore, the three dimensions below are used for the analysis. Each publication in the research series is assessed from the three dimensions so that each research series is presented from the TD research perspective.

- Field—This refers to the environment of research, practical or theoretical (laboratory or desktop). The practical environment can be further detailed in terms of the intensity of interaction between scientists and users. Note that this is not meant for practical or theoretical knowledge, which is more related to the first dimension.
- Disciplinary level—This denotes the level of disciplinarity and is single-, multi-, inter-, or trans-disciplinary 1 [7].
- Outcome knowledge—This dimension has three values: an opportunity identified, a new solution proposed, and a solution enhanced. The second and the third could be merged into one but are kept separate as a proposal to be able to emphasize the newness of a solution relative to extant literature.

Research performed in both theoretical and practical fields was regarded as practical. In addition, only when an article does not propose a solution clearly, was it classified as "an opportunity identified"; otherwise, the research belongs to the "solution" spaces.

#### 3.4.2. Genealogy for PSS Design Research from TD Perspective

The article in focus [21] proposes a new solution that solved two design problems including one with industry. The research "proposing a novel engineering discipline" was conducted in a practical environment. Therefore, it is classified as proposing a new solution, practical, and trans-disciplinary 1. The articles in this genealogy vary both in terms of the outcome and field, as shown in Figure 5, and many link cross space borders. In terms of the disciplinary level, Refs. [36,52] take a similar standpoint as [21]. Research in [36] and onward is deemed as an enhancement of [21] and thus is allocated to the space of a solution enhanced. Ref. [42] concerns interdisciplinary issues, and Ref. [41] discusses "integration of functional modelling in interdisciplinary system development", and therefore may be regarded as interdisciplinary. Identifying the level requires considering the nuance of the contents; for instance, "cross-disciplinary focus of the PSS design approach" [39] may be regarded as interdisciplinary. Ref. [40]

concerns "how different domains can learn from each other and how designers can implement and coordinate them in design activities effectively" and therefore is interdisciplinary.

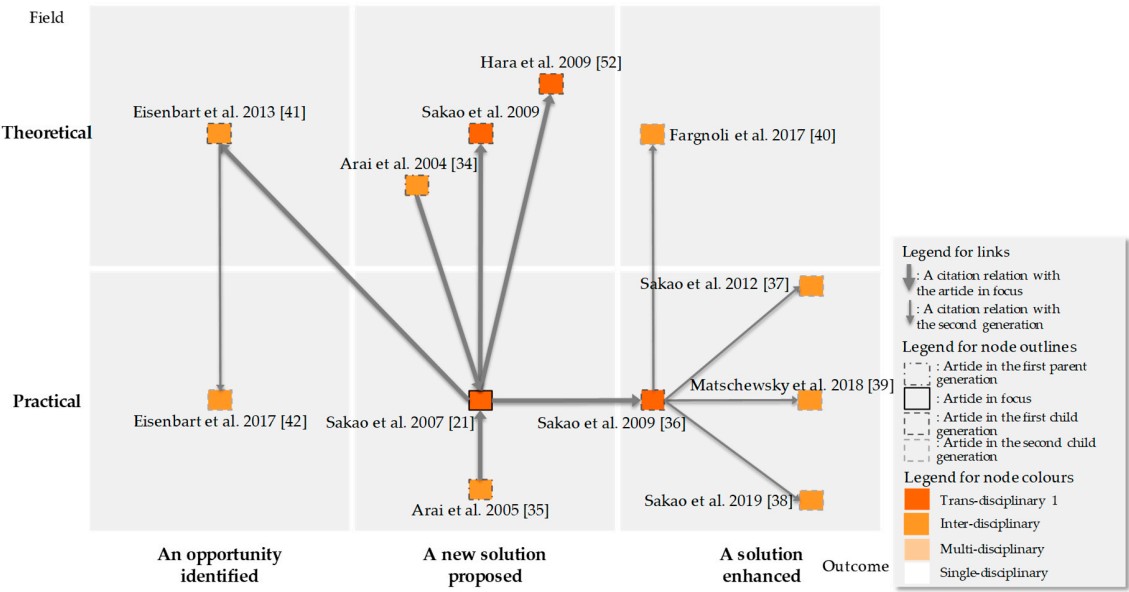

**Figure 5.** Genealogy for PSS design research mapped on the three dimensions: field, disciplinary level and outcome knowledge. Note: the articles in the figure are listed in Appendix C.

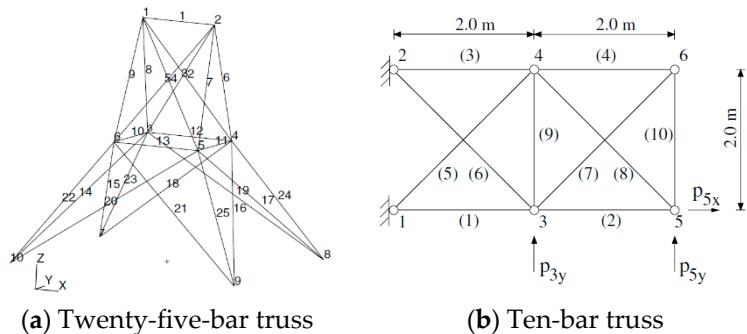

(**a**) Twenty-five-bar truss      (**b**) Ten-bar truss

**Figure 6.** Representations of the two problems in Ref. [32]. Reprinted from Computer Methods in Applied Mechanics and Engineering 193, Doltsinis, I. and Kang, Z., Robust design of structures using optimization methods, 2221–2237, Copyright (2004), with permission from Elsevier.

### 3.4.3. Genealogy for Robust Design Research from TD Perspective

The article in focus [32] presents a research work proposing a new solution solving problems identified originally. The research was conducted in a theoretical environment and may be regarded as single disciplinary. Therefore, it is classified as proposing a new solution, as theoretical, and as single disciplinary. The articles in this genealogy focus on research in a theoretical environment, contributing to improving theories in the mechanical engineering discipline. As shown in Figure 7, the articles are concentrated on the theoretical–solution space. They typically target a benchmark problem, which is an abstraction of a real-world problem. Note that several articles identify original problems and could be classified as opportunity articles but, as stated in Section 3.4.1., those with both a solution and a problem were in principle classified as a solution paper. Some articles present, in addition, more than solving a benchmark problem on a computer. For instance, a hardware experiment with sensors is performed after optimal sensor placement was derived [49]. Further, more concrete problems are modelled: Ref. [53] addressed structure models for hydropower generation. Ref. [54] addressed offshore wind turbines even with "the 5 MW NREL wind turbine model" specifically but is unclear

about the degree of involvement of practitioners. Research work [47] and onward is deemed as an enhancement of [32] and thus allocated to the space of solution enhanced.

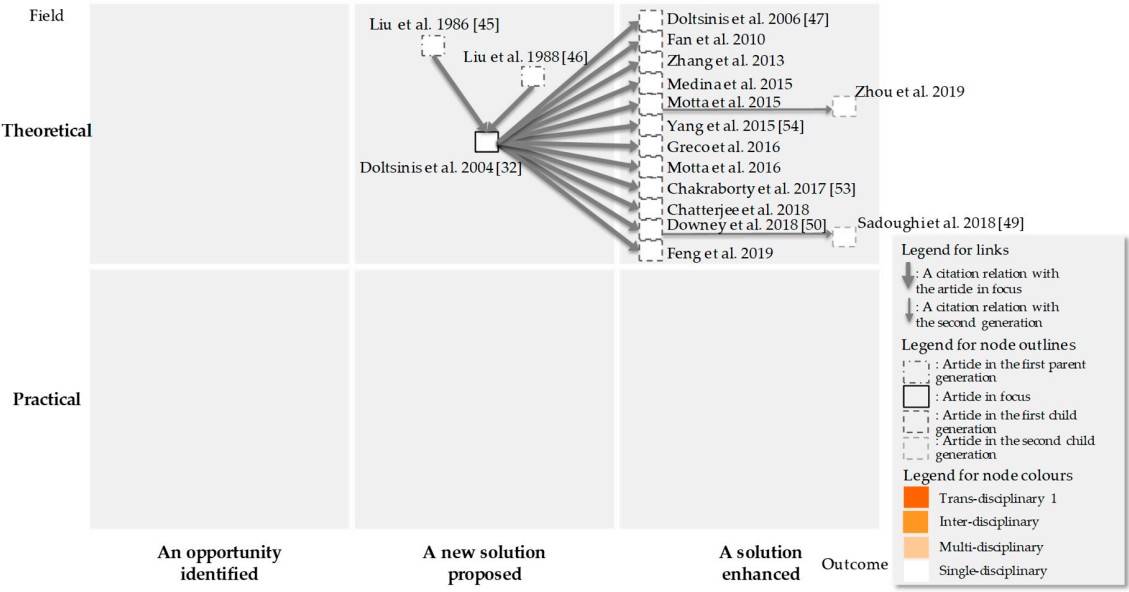

**Figure 7.** Genealogy for robust design research mapped on the three dimensions; field, disciplinary level and outcome knowledge. Note: the articles in the figure are listed in Appendix C.

### 3.4.4. Analysis with Indicators

Adopting indicators for representing the two research series quantitatively is useful for analysis. Here, the indicators below are used for a series of research works to aim at answering the second research question. The higher the first two indicator values are, the stronger the TD feature in the research.

- Frequency of research in a practical field: (the number in the practical field)/(the number of population)
- Frequency of TD-1 research: (the number in TD-1) / (the number of population)
- Frequency of single discipline research: (the number in single-disciplinary)/(the number of population)
- Frequency of a solution provided: ((the number in a new solution proposed) + (the number in a solution enhanced))/(the number of population)

In addition, the indicator value below is of interest to indicate transition trends between practical and theoretical fields (field dimension). This can be calculated by capturing the links in the series of research, which connect two nodes either in the same field or in different fields. The Markov model can be adopted for extending this analysis.

- Probability to switch fields: (the number of links connecting nodes in different fields)/(the number of all the links)

The indicator values of the two genealogies are calculated as in Table 2. The two genealogies are shown in high contrast with each other for the first three indicators and the last: the genealogy of PSS design research has shown significantly higher values than that of robust design research in the first two indicators. This means the former is more TD than the latter. The last indicator, probability to switch fields, received a much higher number in the former, which also supports that the former is more TD. According to the fourth indicator in Table 2, the example PSS design research includes

publications focusing on opportunities (17%), and the example robust design research is published entirely for solutions. Based on the results, these indicators are of potential relevance to assess a series of research works from the TD research perspective.

**Table 2.** Indicator values of the two genealogies [%].

| Indicator | Genealogy of PSS Design Research [21] | Genealogy of robust Design Research [32] |
|---|---|---|
| Frequency of Research in a Practical Field | 58 | 0 |
| Frequency of TD-1 Research | 33 | 0 |
| Frequency of Single Disciplinary Research | 0 | 100 |
| Frequency of a Solution Provided | 83 | 100 |
| Probability to Switch Fields | 55 | 0 |

## 4. Discussion

### 4.1. Effectiveness of Research Series Review (RSR)

The genealogies with their narratives derived in Sections 3.2 and 3.3. show the effectiveness of RSR to capture a series of research that has cause-and-effect relationships thanks to the concept of a gene. It should be emphasized that the outcomes of RSR are substantially focused compared to mere citation analysis: the ratio of articles in the genealogy over those citing the focal article at the first child generation, for instance, was 4% (4/97) and 9% (12/139), as shown in Table 1. These low ratios corroborate the result in a similar review [25], where method usage was focused on a causal relation. This feature of RSR increases the effectiveness and efficiency to understand a research series. The derived genealogies also indicate that the procedure of RSR proposed in Section 2.1. is verified. It should be remarked that judgement of whether a publication includes a gene depends on a threshold to the degree of inclusion. An identical threshold was used within this RSR to maintain consistency (as exemplified in Section 3.2.).

Compared with systematic literature review, RSR has multiple advantages. First, it has potential to be used for the purpose of planning a research series pre-emptively, as well as being used for analysing research works retrospectively. A research series requires more effort by nature compared to a research project, which increases the importance of planning. This advantage makes RSR relevant to research funders as well as researchers, both of whom are involved in planning a research series. As reviewed in Section 1, a practical yet science-based procedure for assessing research planned or performed from the TD research perspective is needed by researchers and funders. Second, RSR can produce results in more depth rather than breadth and clearer traceability on knowledge production within the series of research works that have cause-and-effect relationships. This characteristic is realized by the concept of genes and beneficial for a more focused review, be it retrospectively or pre-emptively.

RSR is carried out based only on citation and content analysis and therefore considered to be useful in practice for eliminating an investigator's mindset and bias in performing a literature review, be it an ad-hoc review or a systematic literature review. In terms of barriers between disciplines, RSR provides the new potential to find publications that would not be found with a predefined keyword query, especially publications using different terms outside the investigator's discipline. Barriers will also be overcome timewise, because RSR focuses on cause-and-effect relations between publications. This is especially relevant to TD research, as it sometimes takes a long time for a research work to reach peak value; see a concrete example for [45] in 1986 and [46] in 1988 having provided a gene for [32] in 2004, as depicted in Figure 4.

### 4.2. Implications for Research on SCP

The example PSS design research series centred with Ref. [21] is one of the earliest research examples that involved successful implementation of a PSS design method impacting working procedures in a

large company having begun in 2012 [39]. Based on the results shown in Sections 3.2 and 3.4.2, this PSS design research series was found to own a TD feature as compared to the robust design research. In addition, how the TD feature was implemented in the research series was documented.

Several aspects of interest can be analysed looking into further details of this genealogy combined with the author's reflection of his own activity in this research. First, research with higher intensity in industry collaboration [36] (reporting "[t]he entire study took approximately five person-months to complete") played a role to boost the following practical research: Refs. [38,39] cited and benefited from [36] rather than the article in focus [21] directly. In fact, the implementation project [39] was granted partly because that company studied Ref. [36], reporting a successful case from a similar sector. This corroborates the importance of the need of high-level contextualization (i.e., consideration beyond laboratory environments) for successful PSS design research, e.g., a company's structure, culture, capabilities and management [24]. Further, the research [38] involving successful PSS cases from a CE perspective on the market in a traditional manufacturing sector has high potential to boost future practical research in a similar setting. The importance of the context and situation of a company is also acknowledged for open innovation [55]. Impact in the opposite direction, i.e., the influence of PSS design on internal and external changes in innovation management, is also reported [56]. On the other hand, this importance of context presents a dilemma for accelerating knowledge advancement. Other research methods than case study, such as survey and statistical data analysis, are effective to gain more quantitative knowledge in PSS research according to Ref. [24]; however, the contextual knowledge is harder to address in those methods.

Second, the pathway from [21] to Ref. [41] and then Ref. [42] in Figure 5 indicates the potential for continuing research in providing a solution in a theoretical field, as they explicitly stated future research opportunities to develop design support (see Section 3.2.). This implies a possibility of research that would realize the pathway "closing" the loop originating from [34] in a figure eight as depicted by Figure 8. In fact, outside the genealogy, a research work is found [57] citing Ref. [42], which proposed a method to explore function vulnerability and identify failure modes in complex systems and then applied it to a case from industry. This means that the authors of [57] developed the method first in a theoretical field and thereby the loop is nearly closed.

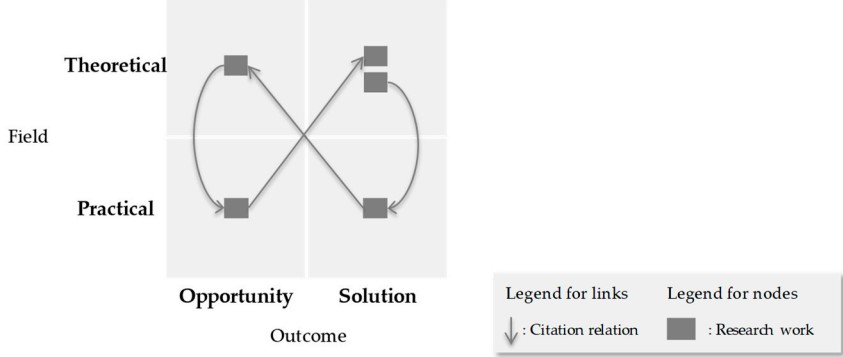

**Figure 8.** A pathway of a research series in a figure eight—"honeybee dancing with inquiry". Note: in general, a pathway may begin at any space in the three-dimensional model and not necessarily the theoretical solution space.

Third, from this successful research series, three lessons learned can be drawn: 1) think big—a major success factor may be the fact that the research [21] began with identifying a grand challenge and need for a TD approach, 2) be patient—there was a gap in the time period for this research series right before the implementation project, but the researcher waited for an opportunity to continue this research until encountering the offer of that project, and 3) act diligently—scientific rigor was maintained throughout the series of research to get academic credit that attracted industry.

Based on the discussion in this section, a successful PSS design research series is hypothesised to form an iterative process between different spaces on the field and outcome dimensions. This hypothesis might be valid even to SCP research or TD research at large. This iteration could be called "honeybee dancing with inquiry" (named after how honeybees move using a "waggle dance"), as depicted by Figure 8; a discussion of the comparison to "dancing with ambiguity" in designers' learning [58] is beyond the scope of this article. This iteration is in clear contrast with the example of robust design research (see Section 3.4.3.), which was found to be carried out mostly in laboratory environments and appears to focus on solutions. This may be because the addressed problems were free of contexts, and therefore, this style of research is suitable. This hypothesis gives some hints to enhance the quality of PSS and SCP research globally, as exemplified further in the next paragraph.

RSR and the following assessment and analysis is expected to contribute to an increase in the quality of a research series as shown by the analysis above (e.g., hypothesis generation), and RSR systematically provides a unit of analysis in such a way that is useful for transdisciplinarity assessment. This is how RSR has potential to contribute to enhance the quality of SCP research and thereby moving our societies to SCP more efficiently. For instance, one may focus too much on a solution when planning research but, thanks to a clearly defined series of research in a longer term and the following assessment, notice poorly defined challenges. This implies the usefulness for educating and training early-stage researchers and practitioners. Even those experienced will be empowered to help a team of multiple people, e.g., share an understanding of a given research series under planning or assessment.

### 4.3. Scientific Newness

RSR was proposed to systematically capture and analyse a series of research works as a unit that has cause-and-effect relationships within itself based on citation content analysis, before being validated with two series of research works in the TD research context. A concept similar to the gene in this article was presented, and a genealogy with no consideration of content analysis was used [19]. In addition, a procedure investigating those papers citing the focal papers and those cited by the focal was proposed [59]. However, RSR as a whole procedure described in this article was not found in the extant literature.

This article also presented one of the first research works that characterized a PSS design research series, as an example of SCP research, in the TD research context, as explained in Section 3.4.2. and as further discussed in Section 4.2. Until now, reviews of PSSs [24,60] reported many instances of research on PSS design having been performed with intense collaboration with practitioners (e.g., [61]) and aiming to transcend from existing disciplines (e.g., [21]). However, virtually no earlier research investigated the characterization of PSS design research on a general level.

## 5. Conclusions and Future Work

### 5.1. Conclusions

Motivated by the need to improve SCP research aiming to make real changes in societies towards environmental sustainability at large, this article proposed a generic method to analyse a research series for transdisciplinarity assessment. This proposal was realized by RSR, which is original to this article, and a systematic review targeting a series of research works. In cases where RSR is applied to publications, it is performed through citation and content analysis with a focus on causal impacts in the TD research context. The proposal was validated with a PSS design research series, a kind of sustainable consumption and production research, compared with a robust design research series, a presumably less TD type of research. Further, the outcomes of the RSR with the two research series were used for presenting indicators of potential relevance from the TD research perspective. A limitation mainly lies in how the content analysis in this research was performed. It was performed by the author, which may risk misunderstanding of the contents through, e.g., overlooking underlying non-stated information that is understandable to an expert in the respective field. However, this risk has been minimized by

the authors of Ref. [32] and the co-author of Ref. [21] having made comments on the descriptions for their respective articles in an earlier draft, and those comments have been incorporated.

In addition, the examined PSS design research series was empirically shown to possess features of TD research, while this type of investigation is absent in the extant literature. In doing so, the importance of high-level contextualization of PSS design research was pointed out. Furthermore, a successful PSS design research series is hypothesised to form an iterative process between theory and practice as well as between problems and solutions.

Lastly, this article is not written to compare the scientific value of TD research and non-TD research, nor can it be used to endorse a claim such as "TD research is always preferred or superior to non-TD research". On many occasions, non-TD research is required and contributes vastly to societies at large. What is needed is the planning and execution of research that is best suited to the needs of the research.

*5.2. Future Work*

5.2.1. Research Series Review (RSR)

RSR aims to capture the activities by scholars publishing their works by building upon extant publications, when applied to publications. These activities are unfortunately expressed unclearly in today's scientific publications in general as implied by the low ratios of publications in the genealogies shown in Section 4.1. This observation corroborates the result from advanced analysis of publications citing Schön's work on reflective practitioners [62]: few instances of citations that engage critically with Schön or build on his ideas were observed and a deeper understanding of citation function was suggested as an interesting and important project. Therefore, further research with RSR is promising. Especially, applying RSR to a series of research works under planning is an important future research work. More case studies with RSR and then consolidation of related procedures (e.g., Ref. [62] adopted a procedure with some commonality to RSR) are suggested in order to eventually enhance the quality of research globally.

In addition, RSR especially provides bibliometric developers with a possibility to go beyond mere citation analysis. It might be expected as an alternative to capture the direct impact of an article on another. This focus on direct impacts is getting more relevant, considering the increasing numbers of references on average per paper and year: about a 3.5% (based on data between 1985 and 2005) and a 1.5% (based on data between 1970 and 1986) increase in psychology [63] and physics [64], respectively. Acknowledging RSR is a time-consuming work if performed thoroughly; (semi)automation partly using locations of mentioning [65] or even natural language processing techniques would be expected to dramatically decrease the time required to capture direct impacts or at least to screen out non-direct impacts.

5.2.2. Dimensions for Assessment from TD Research Perspective

The three dimensions, i.e., field, disciplinary level and outcome knowledge, were used from the TD research perspective in Section 3.4. Other dimensions may be used instead; for instance, they could concern the beneficiary (science/user) [8], the research phase (basic/applied) [66], the region [67], or other. This provides opportunities for future research to develop a framework for better research planning or evaluation from the TD research perspective.

**Supplementary Materials:** The following are available online at http://www.mdpi.com/2071-1050/11/19/5250/s1, Table S1: Generations_PSS_design_research.xlsx, Table S2: Generations_robust_design_research.xlsx.

**Author Contributions:** T.S. has done everything.

**Funding:** This research received no external funding.

**Acknowledgments:** Ioannis Doltsinis and Zhan Kang as the authors of Ref. [32] as well as Yoshiki Shimomura as the co-author of [21] are appreciated for commenting on the descriptions for their respective articles. Discussions during the Transdisciplinary Metrics Workshop hosted by Belmont Forum in the USA in June 2019 that provided the inspiration to enrich some aspects of this paper are acknowledged. Discussions with Jenny Palm of Lund University and Harald Rohracher of Linköping University as well as feedback on an earlier version of this paper given by Sergio Brambila of Linköping University are also appreciated.

**Conflicts of Interest:** The author declares no conflicts of interest.

## Appendix A

The search that yielded the selection of Ref. [32] was performed in Scopus with the TITLE-ABS-KEY being "robust design" AND method AND optimization as well as the article type being "ar". Those hit with this query and have been cited 200-250 times are shown below.

**Table A1.** Candidate articles for robust design research.

| Document Title | Year | Source | Citation |
|---|---|---|---|
| A review of robust optimal design and its application in dynamics | 2005 | Computers and Structures | 247 |
| An integrated framework for optimization under uncertainty using inverse reliability strategy | 2004 | Journal of Mechanical Design | 218 |
| Robust design of structures using optimization methods [32] | 2004 | Computer Methods in Applied Mechanics and Engineering | 206 |
| Nano spray drying: A novel method for preparing protein nanoparticles for protein therapy | 2011 | International Journal of Pharmaceutics | 204 |
| Robust optimization considering tolerances of design variables | 2001 | Computers and Structures | 200 |

## Appendix B

- List of covered journals for the PSS design research (the articles in these journals were all accessible and thus investigated when found in the database).

Advanced Engineering Informatics; Artificial Intelligence for Engineering Design, Analysis and Manufacturing: AIEDAM; Benchmarking; Business Strategy and the Environment; CAD Computer Aided Design; CIRP Annals; CIRP Journal of Manufacturing Science and Technology; Computers and Industrial Engineering; Computers in Industry; Concurrent Engineering Research and Applications; Design Science; Design Studies; Ergonomics; EMJ - Engineering Management Journal; Environmental Innovation and Societal Transitions; Expert Systems with Applications; Future Generation Computer Systems; IEEE Transactions on Services Computing; IFAC-PapersOnLine; International Journal of Advanced Manufacturing Technology; International Journal of Computer Integrated Manufacturing; International Journal of Design Creativity and Innovation; International Journal of Entrepreneurship and Innovation; International Journal of Industrial Engineering : Theory Applications and Practice; International Journal of Mathematics and Computers in Simulation; International Journal of Operations and Production Management; International Journal of Precision Engineering and Manufacturing—Green Technology; International Journal of Production Research; Journal of Cleaner Production; Journal of Computational Design and Engineering; Journal of Engineering and Technology Management—JET-M; Journal of Engineering Design; Journal of Intelligent Manufacturing; Journal of Manufacturing Science and Engineering, Transactions of the ASME; Journal of Manufacturing Technology Management; Journal of Mechanical Engineering Research and Developments; Journal of Open Innovation: Technology, Market, and Complexity; Journal of Reinforced Plastics and Composites;

Journal of Remanufacturing; Journal of Retailing and Consumer Services; Journal of Systems and Software; Journal of Systems Science and Systems Engineering; Kybernetes; Mechatronics; Production Planning and Control; Quality Engineering; Research in Engineering Design; Service Business; Service Industries Journal; Sustainability (Switzerland); Systems Engineering; Tehnicki Vjesnik; Total Quality Management and Business Excellence; TQM Journal; WSEAS Transactions on Business and Economics.

- List of covered journals for the robust design research (the articles in these journals were all accessible and thus investigated when found in the database).

Acta Mechanica Sinica/Lixue Xuebao; Advances in Condensed Matter Physics; Advances in Engineering Software; Advances in Mechanical Engineering; Aeronautical Journal; Applied Mathematical Modelling; Applied Mathematical Modelling; Applied Mathematics and Computation; Applied Soft Computing Journal; Archives of Computational Methods in Engineering; ASCE-ASME Journal of Risk and Uncertainty in Engineering Systems, Part B: Mechanical Engineering; Asian Journal of Civil Engineering; Australian Journal of Electrical and Electronics Engineering; CAD Computer Aided Design; Chinese Journal of Aeronautics; CMES—Computer Modeling in Engineering and Sciences; Composite Structures; Computational Materials Science; Computational Optimization and Applications; Computer Methods in Applied Mechanics and Engineering; Computers and Geotechnics; Computers and Structures; Computers, Materials and Continua; Earthquake Engineering and Engineering Vibration; Energy Conversion and Management; Engineering Optimization; Engineering Structures; Engineering with Computers; European Journal of Operational Research; Finite Elements in Analysis and Design; Frontiers of Mechanical Engineering; Fuzzy Sets and Systems; IEEE Access; International Journal for Numerical Methods in Engineering; International Journal of Advanced Manufacturing Technology; International Journal of Computational Methods in Engineering Science and Mechanics; International Journal of Distributed Sensor Networks; International Journal of Environment and Waste Management; International Journal of Mechanics and Materials in Design; International Journal of Reliability, Quality and Safety Engineering; International Journal of Rotating Machinery; International Journal of Solids and Structures; Journal of Applied Mechanics, Transactions ASME; Journal of Biomechanical Engineering; Journal of Computational Physics; Journal of Engineering (United States); Journal of Fluids and Structures; Journal of GeoEngineering; Journal of Intelligent Manufacturing; Journal of Marine Science and Application; Journal of Mechanical Design, Transactions of the ASME; Journal of Mechanical Engineering Research and Developments; Journal of Mechanical Science and Technology; Journal of Mechanisms and Robotics; Journal of Mechanics of Materials and Structures; Journal of Ocean and Wind Energy; Journal of Optimization Theory and Applications; Journal of Performance of Constructed Facilities; Journal of Sound and Vibration; Journal of Statistical Computation and Simulation; Journal of Systems and Software; Journal of Thermal Stresses; Journal of Vibration and Acoustics, Transactions of the ASME; JVC/Journal of Vibration and Control; KSCE Journal of Civil Engineering; Marine Structures; Mathematical Problems in Engineering; Measurement Science and Technology; Mechanical Systems and Signal Processing; Mechanics and Industry; Mechanics Based Design of Structures and Machines; Multibody System Dynamics; Neural Computing and Applications; Nuclear Technology; Operations Research; Optimization and Engineering; Optimization Letters; Probabilistic Engineering Mechanics; Problems and Perspectives in Management; Proceedings of the Institution of Mechanical Engineers, Part C: Journal of Mechanical Engineering Science; Proceedings of the Institution of Mechanical Engineers, Part G: Journal of Aerospace Engineering; Proceedings of the Institution of Mechanical Engineers, Part O: Journal of Risk and Reliability; Quality and Reliability Engineering International; Reliability Engineering and System Safety; Research Journal of Applied Sciences, Engineering and Technology; Results in Physics; Robotica; Schmerz; Science China Technological Sciences; Sensors (Switzerland); Shock and Vibration; Simulation Modelling Practice and Theory; Smart Materials and Structures; Smart Structures and Systems; Soil Dynamics and Earthquake Engineering; Structural and Multidisciplinary Optimization; Structural Control and Health Monitoring;

Structural Health Monitoring; Structural Safety; Structure and Infrastructure Engineering; Tunnelling and Underground Space Technology; Vehicle System Dynamics; Water Resources Research.

## Appendix C

Articles in the genealogies are listed below; P1, C1, and C2 stand for the first parent generation, the first child generation, and the second child generation. Focus means the article in focus. Below are for the PSS design research in Figure 5.

**Table A2.** Articles in the genealogy for PSS design research.

| Generation | Label | Title | Source |
|---|---|---|---|
| P1 | Arai et al. 2004 [34] | Proposal of Service CAD System—A Tool for Service Engineering | CIRP Annals—Manufacturing Technology |
| P1 | Arai et al. 2005 [35] | Service CAD System—Evaluation and Quantification | CIRP Annals—Manufacturing Technology |
| Focus | Sakao et al. 2007 [21] | A Novel Engineering Discipline for Producers to Increase Value Combining Service and Product | Journal of Cleaner Production |
| C1 | Eisenbart et al. 2013 [41] | An Analysis of Functional Modeling Approaches Across Disciplines | Artificial Intelligence for Engineering Design, Analysis and Manufacturing |
| C1 | Hara et al. 2009 [52] | Service CAD System to Integrate Product Behavior and Service Activity for Total Value | CIRP Journal of Manufacturing Science & Technology |
| C1 | Sakao et al. 2009 [36] | An Effective and Efficient Method to Design Services: Empirical Study for Services by an Investment—Machine Manufacturer | International Journal of Internet Manufacturing and Services |
| C1 | Sakao et al. 2009 | Modeling Design Objects in CAD System for Service/Product Engineering | Computer-Aided Design |
| C2 | Sakao et al. 2012 [37] | A Value-Based Evaluation Method for Product/Service System Using Design Information | CIRP Annals—Manufacturing Technology |
| C2 | Eisenbart et al. 2017 [42] | Taking A Look at the Utilisation of Function Models in Interdisciplinary Design: Insights from 10 Engineering Companies | Research in Engineering Design |
| C2 | Fargnoli et al. 2017 [40] | Uncovering Differences and Similarities among Quality Function Deployment Based Methods in Design for X—Benchmarking in Different Domains | Quality Engineering |
| C2 | Matschewsky et al. 2018 [39] | Designing and Providing Integrated Productservice Systems—Challenges, Opportunities and Solutions Resulting from Prescriptive Approaches in 2 Industrial Companies | International Journal of Production Research |
| C2 | Sakao et al. 2019 [38] | A Methodological Approach for Manufacturers to Enhance Value-in-Use of Service-Based Offerings Considering 3 Dimensions of Sustainability | CIRP Annals—Manufacturing Technology |

The following is for the robust design research in Figure 7.

**Table A3.** Articles in the genealogy for robust design research.

| Generation | Label | Title | Source |
|---|---|---|---|
| P1 | Liu et al. 1986 [45] | A Random Field Finite Elements | International Journal for Numerical Methods in Engineering |
| P1 | Liu et al. 1988 [46] | Transient Probabilistic Systems | Computer Methods in Applied Mechanics and Engineering |
| Focus | Doltsinis et al. 2004 [32] | Robust Design of Structures Using Optimization Methods | Computer Methods in Applied Mechanics and Engineering |
| C1 | Doltsinis et al. 2006 [47] | Perturbation-Based Stochastic FE Analysis and Robust Design of Inelastic Deformation Processes | Computer Methods in Applied Mechanics and Engineering |
| C1 | Fan et al. 2010 | The Robust Optimization for Large-Scale Space Structures Subjected to Thermal Loadings | Journal of Thermal Stresses |
| C1 | Zhang et al. 2013 | Structural Reliability Analysis Based on the Concepts of Entropy, Fractional Moment and Dimensional Reduction Method | Structural Safety |
| C1 | Medina et al. 2015 | Probabilistic Measures for Assessing Appropriateness of Robust Design Optimization Solutions | Structural and Multidisciplinary Optimization |
| C1 | Motta et al. 2015 | Development of A Computational Efficient Tool for Robust Structural Optimization | Engineering Computations |
| C1 | Yang et al. 2015 [54] | Robust Design Optimization of Supporting Structure of Offshore Wind Turbine | Journal of Marine Science and Technology |
| C1 | Greco et al. 2016 | Robust Optimization of Base Isolation Devices under Uncertain Parameters | JVC/Journal of Vibration and Control |
| C1 | Motta et al. 2016 | An Efficient Procedure for Structural Reliability-Based Robust Design Optimization | Structural and Multidisciplinary Optimization |
| C1 | Chakraborty et al. 2017 [53] | A Surrogate Based Multi-Fidelity Approach for Robust Design Optimization | Applied Mathematical Modelling |
| C1 | Chatterjee et al. 2018 | Analytical Moment-Based Approximation for Robust DDesign Optimization | Structural and Multidisciplinary Optimization |
| C1 | Downey et al. 2018 [50] | Optimal Sensor Placement within A Hybrid Dense Sensor Network using An Adaptive Genetic Algorithm with Learning Gene Pool | Structural Health Monitoring |
| C1 | Feng et al. 2019 | An Innovative Estimation of Failure Probability Function Based on Conditional Probability of Parameter Interval and Augmented Failure Probability | Mechanical Systems and Signal Processing |
| C2 | Sadoughi et al. 2018 [49] | Reconstruction of Unidirectional Strain Maps Via Iterative Signal Fusion for Mesoscale Structures Monitored by A Sensing Skin | Mechanical Systems and Signal Processing |
| C2 | Zhou et al. 2019 | An Expanded Sparse Bayesian Learning Method for Polynomial Chaos Expansion | Mechanical Systems and Signal Processing |

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
