# Peer review of "Research Series Review for Transdisciplinarity Assessment—Validation with Sustainable Consumption and Production Research"

_sustainability, doi:10.3390/su11195250_

Round 1
Reviewer 1 Report
The topic is interesting. However, I have some suggestion to advance your article.
Introduction: In this part, it is better to mention the significance of your research. You can explain more about transdisciplinary research and why it is useful in this research topic. What are the advantages of this method in comparison with other research methods?
Research motivation: It would be better to put research gaps and motivations in the Introduction part and rename this section to the theoretical background and explain about the background of the research. For instance sustainable consumption and production, circular economy, and ...
Method and material: In section 3.1.2. the gene has been defined that "a gene is what makes a publication solid, ....". What are the criteria to select the document as a solid publication?
four steps for LSR has been explained: in section 3.2. four criteria have been defined for selecting other articles in focus. What is the reason behind those four criteria? Why the articles between (200-250) citations times have been selected? In this case, you exclude some articles that might have new contributions and important points in this research area. In the third step, How did you select the article by content? Did you read the full text? please explain it more. and simplify this part in a way that is more clear. It is not clear how you choose parents and children?
Results: 4.2.1. it is not so clear and convincing how you select the first and second parents and children. It would be better to explain it in a way that is more clear? If you use keywords to choose the articles, please specify. Did you read the cited articles one by one? If so what are the advantages of this method in comparison to other ways(i.e. manual) to find cause and effect relationship between articles?
Discussion: It would be more coherent to concentrate on a discussion in this part and put further research and limitation in the conclusion section.
Conclusion: As mentioned in the previous section, it is more coherent to put limitations and further research in the conclusion.
Author Response
Author’s responses to comments by the reviewers
The responses are explained below for each comment, while the actual changes on the manuscript are found in the manuscript with the highlighting yellow color. The reviewers’ comments have been constructive, useful and much appreciated. The author has addressed each one of them carefully. Overall, the reviewers have recommended a shorter, crispier and more solid paper. This recommendation makes sense and I followed it. It is hoped that this direction does not turn out to decrease the overall value of the paper.
Reviewer :
The topic is interesting. However, I have some suggestion to advance your article.
Introduction: In this part, it is better to mention the significance of your research. You can explain more about transdisciplinary research and why it is useful in this research topic. What are the advantages of this method in comparison with other research methods?
I highly appreciate your review work and suggestion. The significance is now more elaborated in Lines 76-80 in this section. What is transdisciplinary research is now moved from the old Section 2 to this section. Why it is useful is more elaborated in Lines 73-75 (and also further shown in Section 4). The advantage of this method is explained in Lines 75-76 and more explicitly shown in Section 4.
Research motivation: It would be better to put research gaps and motivations in the Introduction part and rename this section to the theoretical background and explain about the background of the research. For instance sustainable consumption and production, circular economy, and ...
Your recommendation was to move some parts from Section 2 to Section 1 and rename Section 2. The parts to be moved are “2.2.2. Transdisciplinary (TD) research“ (from your previous comment) and “2.2.3. Knowledge gap”. Thank you for giving a sensible alternative. I followed it. In parallel, based on constructive comments by other multiple reviewers, the third contribution in the original manuscript (characterizing a PSS design research series) is removed, and thus “2.2.1. PSS research” has no strong reason to take the space here. These changes have led to no need of keeping this section as a separate one. Therefore, how to name it is not an issue anymore.
Method and material: In section 3.1.2. the gene has been defined that "a gene is what makes a publication solid, ....". What are the criteria to select the document as a solid publication?
The previous manuscript did not touch upon how to “select the document as a solid publication”. It described how to identify a published research article in a series or research works. Therefore, this question is understood as caused by miscommunication. To avoid this miscommunication, the quoted expression is now removed.
four steps for LSR has been explained: in section 3.2. four criteria have been defined for selecting other articles in focus. What is the reason behind those four criteria? Why the articles between (200-250) citations times have been selected? In this case, you exclude some articles that might have new contributions and important points in this research area. In the third step, How did you select the article by content? Did you read the full text? please explain it more. and simplify this part in a way that is more clear. It is not clear how you choose parents and children?
The reasons behind the four criteria are now elaborated in Lines 169-176. Regarding “you exclude some articles that might have new contributions and important points in this research area”, there seems miscommunication. The author’s intention has not been to cover most contributions in the area. To avoid this miscommunication, a note has been added in Lines 178-180. The selection process is now elaborated in Lines 177-175. Assuming I understand your comment, “and simplify this part in a way that is more clear”, in a correct way, the revised description is regarded as clear enough. Regarding “how you choose parents and children?”, this process was described in a generic manner in the method section (now, labeled as Section 2.1). Seeing this question raised by none of the three other reviewers, this process might be regarded as clear in the previous manuscript. Nonetheless, to avoid this question by a reader, “resulting in a set of articles for each generation (the first child and parent generations and so forth)” is now added in Lines 118-119. If the chosen numbers of parent and child generations is meant by this question, to avoid this question by a reader, a new statement is now added in Lines 106-107; “discussing how many generations are suitable for LSR is beyond this article”.
Results: 4.2.1. it is not so clear and convincing how you select the first and second parents and children. It would be better to explain it in a way that is more clear? If you use keywords to choose the articles, please specify.
The question about “how you select the first and second parents and children” is already answered in the response above.
Did you read the cited articles one by one? If so what are the advantages of this method in comparison to other ways(i.e. manual) to find cause and effect relationship between articles?
As was stated in the previous manuscript, “The total of the numbers in Table 1 is 546, indicating the number of investigations of citation content analysis (besides the two articles in focus). This means that the accuracy of semantic analysis for citation explained in Section 3.1.3 was ensured for the 546 articles and not each article was entirely read.” If each of the 546 articles was read (sentence by sentence) is not an issue for LSR (now named as RSR), as many parts of an article are irrelevant for concerned genes. Each of the 546 articles was investigated according to the proposed procedure. Because it was not necessarily read, the question on advantages is not relevant now but is in more general elaborated in Lines 422-432.
Discussion: It would be more coherent to concentrate on a discussion in this part and put further research and limitation in the conclusion section.
Thank you for giving a sensible alternative. I followed it.
Conclusion: As mentioned in the previous section, it is more coherent to put limitations and further research in the conclusion..
I followed your suggestion.
Lastly, you indicated “Moderate English changes required”. However, no more information regarding this is provided in your report. In addition, all the other three reviewers indicate “English language and style are fine/minor spell check required”. Further, the original manuscript had been polished by a professional native science writer with a PhD degree in production. Therefore, the language quality of the original manuscript is regarded as sufficiently high and the language editing remains at the level of minor spell check at least in this revision.
Reviewer 2 Report
Interesting manuscript. However, the approach appears difficult to follow since several topics are attempted simultaneously.
(i) "This article proposes an innovative way to assess the transdisciplinarity of a research series in qualitative and quantitative terms."
To my understanding, innovative way is "What is suggested to be called a literature series review (LSR) follows the concept and merit of a systematic literature review.
1. Identify an article and its genes, 2. Arrange articles in generations, 3. Capture a genealogy centred by the article, 4. Derive a narrative for the genealogy"
How does this model contribute to CE and SCP?
(ii) Please explicitly mention the literature that highlights potential of PSS to CE and SCP. In the review, I was unable to find study that makes a similar claim to following:
"As reviewed above, the literature has shown the great potential of PSSs to contribute to the CE and SCP, but their potential is yet to be realized in our societies."
(iii) Typo errors present
e.g. repeatedly abbriviated: "literature series review (LSR)", few others also
(iv) Difficult to follow how the methodology helped arriving at the conclusions. e.g. "this article filled the gap of knowledge for funders and researchers to plan, facilitate, and evaluate TD research for PSSs and SCP at large"
(v) Consider to cutting it short and making it more explicit/crisp for the reader
Author Response
Author’s responses to comments by the reviewers
The responses are explained below for each comment, while the actual changes on the manuscript are found in the manuscript with the highlighting yellow color. The reviewers’ comments have been constructive, useful and much appreciated. The author has addressed each one of them carefully. Overall, the reviewers have recommended a shorter, crispier and more solid paper. This recommendation makes sense and I followed it. It is hoped that this direction does not turn out to decrease the overall value of the paper.
Reviewer :
Interesting manuscript. However, the approach appears difficult to follow since several topics are attempted simultaneously.
I am glad that you see this as an interesting manuscript. I highly appreciate your review work and insightful comments.
Based on your comments and the other reviewers’ comments, I revised this paper to focus on the proposal of LSR (now named as RSR) and related discussion on SCP and TD. Substantial texts and figures (e.g., Figure 4) are removed. It is indicated in Section 1. Now, thanks to the simplicity, Figure 1 is also removed.
(i) "This article proposes an innovative way to assess the transdisciplinarity of a research series in qualitative and quantitative terms."
To my understanding, innovative way is "What is suggested to be called a literature series review (LSR) follows the concept and merit of a systematic literature review.
Identify an article and its genes, 2. Arrange articles in generations, 3. Capture a genealogy centred by the article, 4. Derive a narrative for the genealogy"
How does this model contribute to CE and SCP?
The contribution to SCP is now elaborated in the last paragraph of Section 4.2. SCP “also serves as a basis for facilitating the transition of societies towards a circular economy (CE)” as stated in Section 1.
(ii) Please explicitly mention the literature that highlights potential of PSS to CE and SCP. In the review, I was unable to find study that makes a similar claim to following:
"As reviewed above, the literature has shown the great potential of PSSs to contribute to the CE and SCP, but their potential is yet to be realized in our societies."
Now, discussion of PSS, as one of the three contributions in the original manuscript is removed by following the reviewers’ recommendations. Therefore, the quoted text is removed and there is now no need to further statements.
(iii) Typo errors present
e.g. repeatedly abbriviated: "literature series review (LSR)", few others also
In the original manuscript, I intentionally did so because some repetition helps a reader. But, no repetition is also a sensible way, which I followed now.
(iv) Difficult to follow how the methodology helped arriving at the conclusions. e.g. "this article filled the gap of knowledge for funders and researchers to plan, facilitate, and evaluate TD research for PSSs and SCP at large"
Thank you for your careful review. The contribution to SCP is now elaborated in the last paragraph of Section 4.2 and thereby the conclusion is now better supported.
(v) Consider to cutting it short and making it more explicit/crisp for the reader.
Thank you so much for your advice. I find it coherent with the comment by Reviewer 3 according to my interpretation of your advice. Therefore, the contribution of the revised manuscript focuses on LSR (now RSR).
Reviewer 3 Report
Dear author,
I absolutely appreciate your endeavor to contribute to transdisciplinarity research and sustainability research and circularity research and to research methodology. So you strive to capture very many topics and this is the problem of your article: It is a bit puzzling as you do not really state what the focus of your work is - it is a bit like a cafeteria where everyone can pick from what he/she wants. Maybe it you might skip the sustainability and at any rate the circularity connections. And you should try to phrase 2 or 3research questions that you concentrate on. Then the paper might get a clear concept and line of argument.
As for your newly developed method, I do not understand its value add. Does not a broader variety of literature approaches also broaden the the view of a problem tackled. Your genealogy approach seems interesting, but it is of academic interest, I cannot see (from the point of transdisciplinarity) the practical value.
Which brings me to my next point: Calling the paper work on tansdisciplinarity involves a broad discussion of the concept, not just a quote that says: That is it! Fullstop. There are several approaches to this concept that should be theoretically discussed and can (only) then be made the basis to what you call your OFD model. This model must be supported much more by theoretical inputs, reasoning, and others. The model as it is here, is irreproducible, arbitrary and at haphazard. This must be redone and amended completely! For its sake, I would personally skip the new method... Furthermore, it does not become clear, why you chose the two concepts (why two? Why these? And why these two publications, not others?)
Actually the paper provides so very nice ideas, but is technically not well done.
Author Response
Author’s responses to comments by the reviewers
The responses are explained below for each comment, while the actual changes on the manuscript are found in the manuscript with the highlighting yellow color. The reviewers’ comments have been constructive, useful and much appreciated. The author has addressed each one of them carefully. Overall, the reviewers have recommended a shorter, crispier and more solid paper. This recommendation makes sense and I followed it. It is hoped that this direction does not turn out to decrease the overall value of the paper.
Reviewer :
Dear author,
I absolutely appreciate your endeavor to contribute to transdisciplinarity research and sustainability research and circularity research and to research methodology. So you strive to capture very many topics and this is the problem of your article: It is a bit puzzling as you do not really state what the focus of your work is - it is a bit like a cafeteria where everyone can pick from what he/she wants. Maybe it you might skip the sustainability and at any rate the circularity connections. And you should try to phrase 2 or 3research questions that you concentrate on. Then the paper might get a clear concept and line of argument.
Dear Reviewer,
I am glad that you appreciate my work. I highly appreciate your review work and insightful comments.
Based on your and the other reviewers’ comments, I revised this paper to focus on the proposal of LSR (now named as RSR) and related discussion on SCP and TD. It is indicated in Section 1. Two research questions are formulated as well.
As for your newly developed method, I do not understand its value add. Does not a broader variety of literature approaches also broaden the the view of a problem tackled. Your genealogy approach seems interesting, but it is of academic interest, I cannot see (from the point of transdisciplinarity) the practical value.
Thank you for your careful review. The added value is now explicitly discussed in the second (and partly the third) paragraph of Section 4.1. During this revision, it was made clear that one of the added values is its applicability to research, be it under planning or already completed. Therefore, “literature series review (LSR)” is now renamed to “research series review (RSR)”.
Which brings me to my next point: Calling the paper work on tansdisciplinarity involves a broad discussion of the concept, not just a quote that says: That is it! Fullstop. There are several approaches to this concept that should be theoretically discussed and can (only) then be made the basis to what you call your OFD model. This model must be supported much more by theoretical inputs, reasoning, and others. The model as it is here, is irreproducible, arbitrary and at haphazard. This must be redone and amended completely! For its sake, I would personally skip the new method... Furthermore, it does not become clear, why you chose the two concepts (why two? Why these? And why these two publications, not others?)
Thank you so much for your insightful comment on the OFD model. Proposing the OFD model with sufficient scientific reasoning was highly challenging with keeping the conciseness of the original manuscript. Now, I find your suggestion sensible concerning this issue. Therefore, the OFD model is not proposed in a scientific sense in this revised manuscript (Figure 4 is now removed) but only used as a way of representation of the results from the RSR. In addition, the OFD model is touched upon as a possible topic for interesting and relevant future work. Nonetheless, the rational for the three dimensions is described in Section 3.4.1.
Accordingly, the title of this article is now adjusted.
Regarding the choice in the last sentence of your comment, I understand you refer to the literature series. Now, the focus of this paper is on LSR and the third contribution in the original manuscript (characterizing a PSS design research series) is removed. Therefore, the importance of the choice is substantially decreased. Nevertheless, the rational of the choice is more clearly described in Section 2.2.
Actually the paper provides so very nice ideas, but is technically not well done.:
Thank you for clarifying the point to be improved. The technical aspect is improved as explained above.
Reviewer 4 Report
In this article, the author claims that transdisciplinary (TD) research is a promising way to enhance PSS research and hence proposes a way to assess the transdisiciplinarity of a research series in a qualitative and quantitative terms.
The author might find TD research relevant to enhance PSS research, it is not really clear from the introduction how that would happen. The motivation for doing TD research is driven more from the funding agencies than anybody else.
The definition used for PSS is very old and does not include the digital aspect of the modern era. We are moving towards smart PSS and author hardly discusses the transition in the whole article.
At first,literature series review (LSR) seems to be an innovative way. But how is it really better than a regular systematic review. What is the real added value? Two articles were given importance based on the citations. One of them is the author's previous article which is 12 years old and the other is 15 years old. Why should these two articles get emphasis? I think these two hardly represent the modern picture.
I believe the LSR could be explained further and more modern articles should be emphasized.
Author Response
Author’s responses to comments by the reviewers
The responses are explained below for each comment, while the actual changes on the manuscript are found in the manuscript with the highlighting yellow color. The reviewers’ comments have been constructive, useful and much appreciated. The author has addressed each one of them carefully. Overall, the reviewers have recommended a shorter, crispier and more solid paper. This recommendation makes sense and I followed it. It is hoped that this direction does not turn out to decrease the overall value of the paper.
Reviewer :
In this article, the author claims that transdisciplinary (TD) research is a promising way to enhance PSS research and hence proposes a way to assess the transdisiciplinarity of a research series in a qualitative and quantitative terms.
The author might find TD research relevant to enhance PSS research, it is not really clear from the introduction how that would happen. The motivation for doing TD research is driven more from the funding agencies than anybody else.
Now, discussion of PSS, as one of the three contributions in the original manuscript is removed by following the reviewers’ recommendations. How TD research contributes to SCP research instead is now explained in addition in the second paragraph of Section 1 and the last paragraph of Section 4.2. The addition includes usefulness for researchers: see the last paragraph of Section 1. Further, it is discussed in the second paragraph of Section 4.1, too: TD research can be planned by researchers, too.
The definition used for PSS is very old and does not include the digital aspect of the modern era. We are moving towards smart PSS and author hardly discusses the transition in the whole article.
Now, the focus of this paper is on LSR (now named as RSR) and the third contribution in the original manuscript (characterizing a PSS design research series) is removed. Therefore, PSS is not a major subject of this paper but so is SCP. Therefore, it is not so relevant any more to discuss more about PSS in addition.
At first,literature series review (LSR) seems to be an innovative way. But how is it really better than a regular systematic review. What is the real added value? Two articles were given importance based on the citations. One of them is the author's previous article which is 12 years old and the other is 15 years old. Why should these two articles get emphasis? I think these two hardly represent the modern picture.
Thank you for your careful review. The added value is now explicitly discussed in the second (and partly the third) paragraph of Section 4.1.
Now, the focus of this paper is on LSR and the third contribution in the original manuscript (characterizing a PSS design research series) is removed. Therefore, the importance of the choice is substantially not so high. Nonetheless, the choice is explained more clearly now. In addition, “It should be noted that these two articles are examples used for testing RSR and there is no intention to argue Ref. [18] as a representation of SCP research” is added now. Thereby, the risk of misunderstanding that the example research series are understood as representative is minimized.
I believe the LSR could be explained further and more modern articles should be emphasized.
RSR is now further explained in Section 2.1. Assuming “more modern articles” are meant for those on PSS, now the issue is not relevant anymore, because the third contribution in the original submission is now removed.
Round 2
Reviewer 1 Report
In compare to the previous version, the current version is improved and can be published.
Author Response
Thank you for your review. No action was needed.
Reviewer 2 Report
Conceptually the article is fine. A few typo and editing errors exist.
Author Response
Thank you for your review. I found an error concerning acronyms and corrected it (spelling out PSS when it appeared first).
Reviewer 3 Report
Dear author,
I want to express my congratulations on the improvement in scientific clearness! Unfortunately your new scientific approach misses to link sustainable production and consumption to the product service systems that you still use as a research object. PSS are not part of the research questions, so it is questionable why you address so much effort into them. Accordingly you must elaborate on that!
And again I would suggest to delve into transdisciplinarity a bit deeper, as this is your main argument for the new method!
Anyway, I can see the progress and wish you well for finishing the article!
Author Response
Dear author,
I want to express my congratulations on the improvement in scientific clearness! Unfortunately your new scientific approach misses to link sustainable production and consumption to the product service systems that you still use as a research object. PSS are not part of the research questions, so it is questionable why you address so much effort into them. Accordingly you must elaborate on that!
Dear Reviewer,
I am glad that you acknowledge improvement of the manuscript. I highly appreciate your review work and insightful comments. Based on your comments, I have now elaborated on the motivation in Section 2.2.
And again I would suggest to delve into transdisciplinarity a bit deeper, as this is your main argument for the new method!
Anyway, I can see the progress and wish you well for finishing the article!
I elaborated on the need of a new method for TD research by explaining its features, characteristics and structure of TD research in Section 1.
Reviewer 4 Report
The author has done a good job with some clarification of concepts. I accept this current version.
Author Response

(The authors gave the same response as above.)

Round 3
Reviewer 3 Report
Dear Authhor,
what an improvement from the first version! I can now happily support the publication of the paper.
All the best on your further scientific journey!